# Just say 'I don't know': Understanding information stagnation during a highly ambiguous visual search task

**Hayward J. Godwin** [1] *, **Michael C. Hout** [2]

**1** School of Psychology, University of Southampton, Southampton, Hampshire, United Kingdom,
**2** Department of Psychology, New Mexico State University, Las Cruces, New Mexico, United States of America

* hayward.godwin@soton.ac.uk

## Abstract

Visual search experiments typically involve participants searching simple displays with two potential response options: 'present' or 'absent'. Here we examined search behavior and decision-making when participants were tasked with searching ambiguous displays whilst also being given a third response option: 'I don't know'. Participants searched for a simple target (the letter 'o') amongst other letters in the displays. We made the target difficult to detect by increasing the degree to which letters overlapped in the displays. The results showed that as overlap increased, participants were more likely to respond 'I don't know', as expected. RT analyses demonstrated that 'I don't know' responses occurred at a later time than 'present' responses (but before 'absent' responses) when the overlap was low. By contrast, when the overlap was high, 'I don't know' responses occurred very rapidly. We discuss the implications of our findings for current models and theories in terms of what we refer to as 'information stagnation' during visual search.

## Introduction

Suppose that you are searching for an important document in your home: you know that it is *somewhere* in your home, but you are not sure where it might be. After much searching, your efforts are fruitless. In this situation, you might assume that you *don't know* where the document is rather than conceding that it has been thrown away. Perhaps you take a break from the search and will try again later; perhaps you will wait to ask a housemate or a partner for any suggestions for where it could be. In this example, reaching an 'I don't know' decision serves as an intermediate step between finding a target and accepting that you have exhausted all possible avenues to detect it.

It is these 'I don't know' decisions that we examine here by engaging participants in a visual search task wherein the target is very difficult to detect. Such 'I don't know' decisions are not just common in mundane searches of our homes: they are also common in real-world search tasks as well, such as those in radiology and airport baggage screening. Difficult-to-resolve searches are also commonplace in other forms of everyday searches, such as information

**Competing interests:** The authors have declared that no competing interests exist.

searches (e.g., seeking to answer a question by looking for information on a webpage). To take the example of airport baggage screening, airport screeners are often given the opportunity to respond that they 'Do Not Know' what a particular object is (that is, they cannot be certain it is safe or not). However, it is important to note that this is not the case for every airport in every country: the rules and regulations vary from airport to airport, from country to country, region to region, and from one baggage screening system to another. Likewise, in complex and ambiguous radiological searches, there may be occasions where radiographers are unable to categorically determine whether a target is present or absent. There are a number of different reasons why an 'I don't know' response might be given. Perhaps a searcher has reached one particular object that they are uncertain is a target or a non-target; or perhaps the searcher is simply faced with examining such an ambiguous display that it is impossible for them to inter-pret it in a meaningful manner see also [1].

Standard visual search experiments only give participants two response options: 'present' or 'absent'. These experiments typically involve participants being asked to search for a single, simple target object that is easily detected amongst a set of non-overlapping, simple distractors (e.g., participants may be asked to search for a *T* shape amongst *L* shapes). Because the stimuli in visual search experiments are typically so simple and easily identified, it would not make sense to provide participants an 'I don't know' response option–indeed, it is only when highly ambiguous stimuli are used that such an option would be necessary.

From a theoretical perspective, the current project's examination of 'I don't know' responses in visual search is important in expanding existing models and theories of search. To our knowledge, no prior studies of visual search have given participants an 'I don't know' response option, and because of this, current models and theories do not seek to account for how such responses are reached by searchers. Clearly this is an important gap to fill given the frequency with which human observers may be uncertain as to the presence or absence of their target (e.g., "have I really searched everywhere?"), or may be uncertain regarding whether or not inspected material is in fact unambiguously the target that they were searching for (e.g., "is this bit of tissue a tumor?").

Here, we have turned to the prominent *Guided Search* model [2] as a starting point for our investigation. We have used this model as a starting point because the manner in which the model is structured is common to virtually all models of visual search. That is, most (if not all) models of search seek to capture the process by which objects are examined, targets are detected, and target-absent responses are generated [3–8]. This model, and many others, con-ceptualize search as involving a series of questions being 'asked' by the visual search system see also [9]. We sketch this out in Fig 1, using example stimuli from our experiment, wherein par-ticipants searched for the letter 'o' embedded amongst other overlapping letters. The top panel of Fig 1 depicts the examination of a set of letters that do not contain a target. Here, the clusters are examined in turn, and the first question that is asked is: does this cluster contain the target? In *Guided Search*, this is treated as a diffusion process towards one of two thresholds: target or distractor. If the answer to this first question is 'yes,' a 'present' response is given; if the answer is 'no,' there is then a check to determine whether the trial should be terminated (or if search should continue by examining other clusters). Search termination is governed here by an internal timer. That timer is based upon past experience and is reduced following a correct response (or increases after a target has been missed) [10]. If the timer's threshold has not been reached, search proceeds; if the threshold has been reached, then an 'absent' response is gener-ated seeing that no target has yet been found. In the example of the top panel of Fig 1, the timer has not yet been reached, so search continues.

Our simple approach here was to assume that 'I don't know' responses will be generated when searchers are unable to answer the first of these two questions. We sketch this out in the

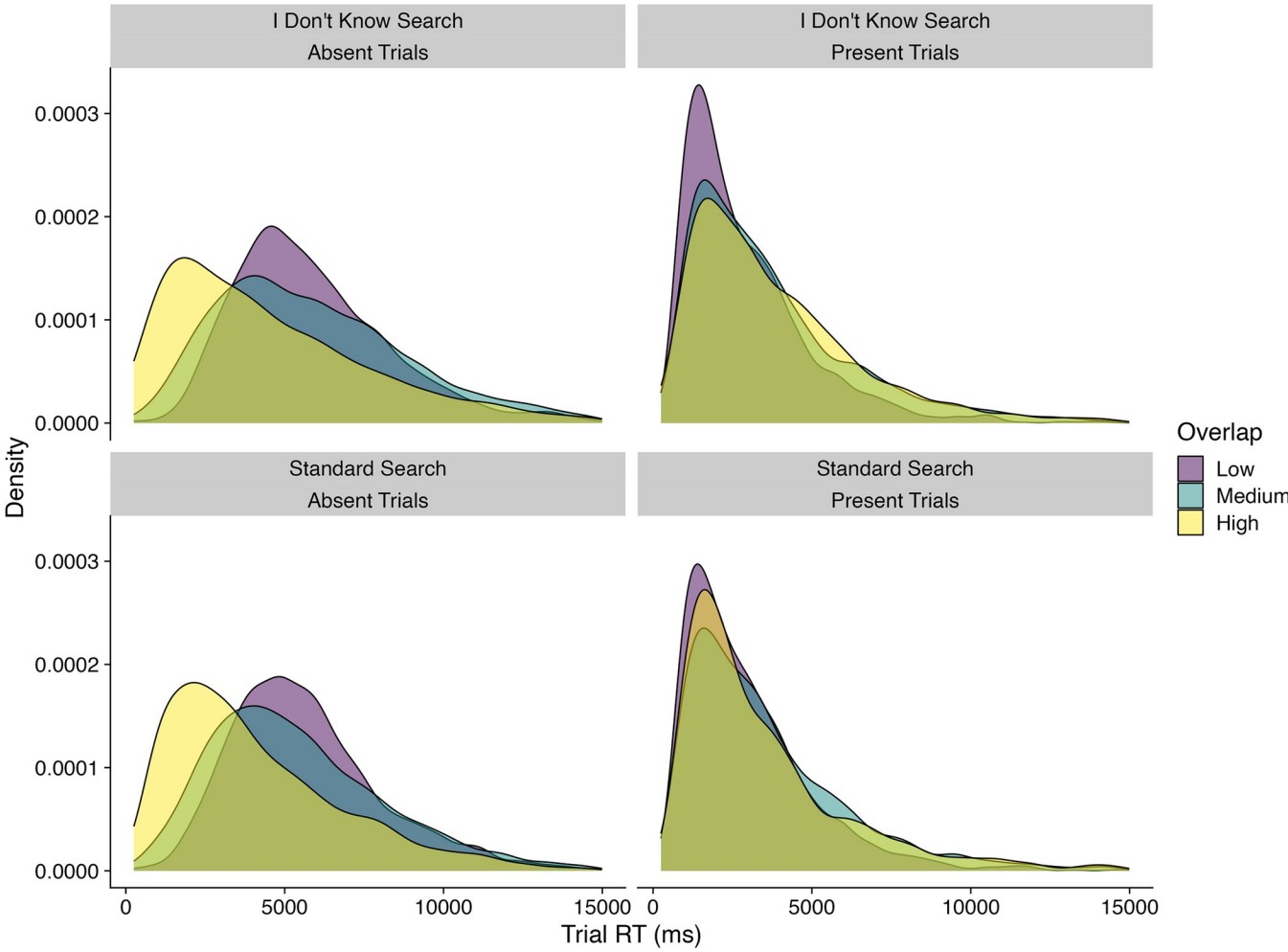

**Fig 1. Decision-making in our visual search task viewed in terms of the guided search model.**

lower panel of Fig 1. Here, in this example, a cluster of objects is being examined but it is very difficult to determine whether an 'o' is present within that cluster. The diffusion process here in the lower panel of Fig 1 remains flat for some time, with there being insufficient visual information available to either confirm the presence of a target or reject all objects in the cluster as distractors. Put another way, this can be regarded as the diffusion process reaching what we call an information 'stagnation' or 'stalemate'.

Our assumption is that, once this information stagnation has occurred for a sufficient period of time, an 'I don't know' response will be generated. A simple set of predictions emerge from this assumption, but before describing them, we will briefly sketch out the experimental design that we used in order to better detail our predictions. As noted above, in our study, participants searched for the letter 'o' embedded amongst other letters in the displays. In the *Standard Search* condition, participants were asked to respond either 'present' or 'absent'; in the *I Don't Know Search* condition, participants could respond 'present', 'absent' or 'I don't know'. We varied the level of overlap of objects (low, medium, high) in the displays to increase the likelihood that participants would be faced with clusters of objects in which it was very difficult–if not impossible–to determine the presence or absence of a target.

We made the following predictions. First, we predicted that in both search conditions, response accuracy would reduce as overlap increased, consistent with previous studies that have examined overlap in search [11]. Second, we predicted that Reaction Times (RTs) would increase as overlap increased, though the magnitude of this we expected to be quite small, in line with previous research that examined response accuracy and RTs as overlap increased (Godwin et al., 2017). Third, we predicted that the proportion of 'I don't know' responses would increase as overlap increased, effectively reducing the proportions of target-present and target-absent trials respectively. Fourth, we predicted that response accuracy would be higher for the standard search condition than the I don't know search condition. This prediction was made because we expected participants in the I don't know search condition to be often responding 'I don't know' rather than being forced to make a guess regarding target present or absence (as in the standard search condition).

Finally, we predicted that RTs for 'I don't know' responses would be faster than 'absent' responses but slower than 'present' responses for the lower levels of overlap only. Our final prediction here requires more detailed explanation and was based upon the following logic: In a purely exhaustive self-terminating search, during target-absent trials, all objects are examined before a response is made. When a participant is faced with a target-present trial, the target will be found after examining, on average, half of the objects in a display. For this reason, target-absent RTs are typically double those of target-present RTs [12]. All of this holds when targets can easily be differentiated from distractors, *and* when only target-present and target-absent responses can be provided by participants.

However, what happens when some targets cannot be differentiated from distractors? Put within the context of our study, we can assume that for every cluster of objects examined, there is a probability that the cluster being examined will generate an 'I don't know' response. In the simplest case, the trial will end when such a cluster is examined. As the overlap increases, more of these 'I don't know' response clusters will be presented to participants in the displays, and as such, the likelihood of a rapid 'I don't know' response increases. Because of this, we expected that target-present RTs would be faster than 'I don't know' responses when there were relatively few very difficult clusters per display (i.e., in the lower overlap conditions). By contrast, we predicted that this effect would be reversed in the higher overlap displays, with 'I don't know' RTs becoming faster than target-present RTs when overlap made the task very difficult because, in such displays, it would be highly likely that the participants would rapidly encounter a cluster that would generate an 'I don't know' response. Of course, it is possible that more complex scenarios and strategies could be used by participants. For example, they may decide to provide an 'I don't know' response based on an overall impression of how difficult a display appears to be, or to provide an 'I don't know' response after encountering several very difficult clusters. For now, given that this is the first study to examine 'I don't know' responses when searching ambiguous displays, we are focusing on the simplest possible case.

## Method

### Ethical approval

Ethical approval for the study was gained from the Psychology Ethics Committee at the University of Southampton (Study ID: ERGO 62661, Date of Approval: 19/1/2021).

### Participants

Participants took part in the study as part of an undergraduate research methods module at the University of Southampton. Consent was obtained by participants completing an online

consent form. Two cohorts of students took part in January-February of 2021 and January-February of 2022. A total of 277 participants completed the study and consented for their data to be used for the purposes of research.

We did not conduct a power analysis in advance given a study of this nature has not been completed previously. That being said, we sought to recruit at a far higher rate than previous studies of overlapping displays. For example, a recent study that manipulated levels of overlap in visual search in a similar manner to how we manipulated overlap here [11] recruited 32 participants for different stimulus types, and we far exceeded this sample size.

Only one of the authors (HG) had access to information that could be used to identify the students (their student ID numbers). These numbers have been removed from the dataset shared online and were only used to check how many of the students in the class took part in the study.

## Apparatus

Participants took part in the study using their own computers and laptops. Data were collected using an in-house data collection server at the University of Southampton. This server uses software called *Just Another Tool for Online Studies* [13] to deliver the study to participants and record the datasets. The study itself was programmed using jsPsych [14]. This is a Java-Script library that has been demonstrated to have a high level of temporal precision [15]. In addition, we used the jsPsychophysics plugin for jsPsych, which has also been shown to have a very high level of temporal precision [16].

Participants responded 'present' using the 'm' key, 'absent' using the 'z' key and (where available) 'I don't know' using the spacebar.

## Design and procedure

Participants were randomly assigned to one of two search conditions: *Standard Search*, and *I Don't Know Search*. In the Standard Search condition, participants were given two responses options–'present' or 'absent'. In the I Don't Know Search condition, participants could respond 'present', 'absent', or 'I don't know'. Participants in the Standard Search condition were given the following instructions: "If you think you can see a letter o in the display, please press the **'m'** key. If you think the letter o is not present in the display, please press the **'z'** key." Participants in the I Don't Know Search condition were given the following instructions: "If you think you can see a letter o in the display, please press the **'m'** key. If you think the letter o is not present in the display, please press the **'z'** key. If you do not know whether a target is present, please press **Space.**" We kept the instructions here minimal particularly for the I Don't Know Search condition in order to avoid being overly prescriptive regarding *how* participants should reach the 'I don't know' response. Put simply, we did not want to create demand characteristics regarding how that response should be generated.

After giving their consent to take part, participants were given information about the task and then completed 12 practice trials. After this, the experimental trials began. There were a total of 270 experimental trials. Trial order was randomized for each participant. A target was presented on 50% of trials. We varied the difficulty of the displays using three different levels of overlap (low, medium, high) with an equal number of trials for each level of overlap.

Each trial began with the presentation of a fixation cross for 500 ms. After this, the search display appeared and remained visible until a response was given. Following each response, the display was blank for 500 ms. In Fig 2 we present an overview of the trial sequence.

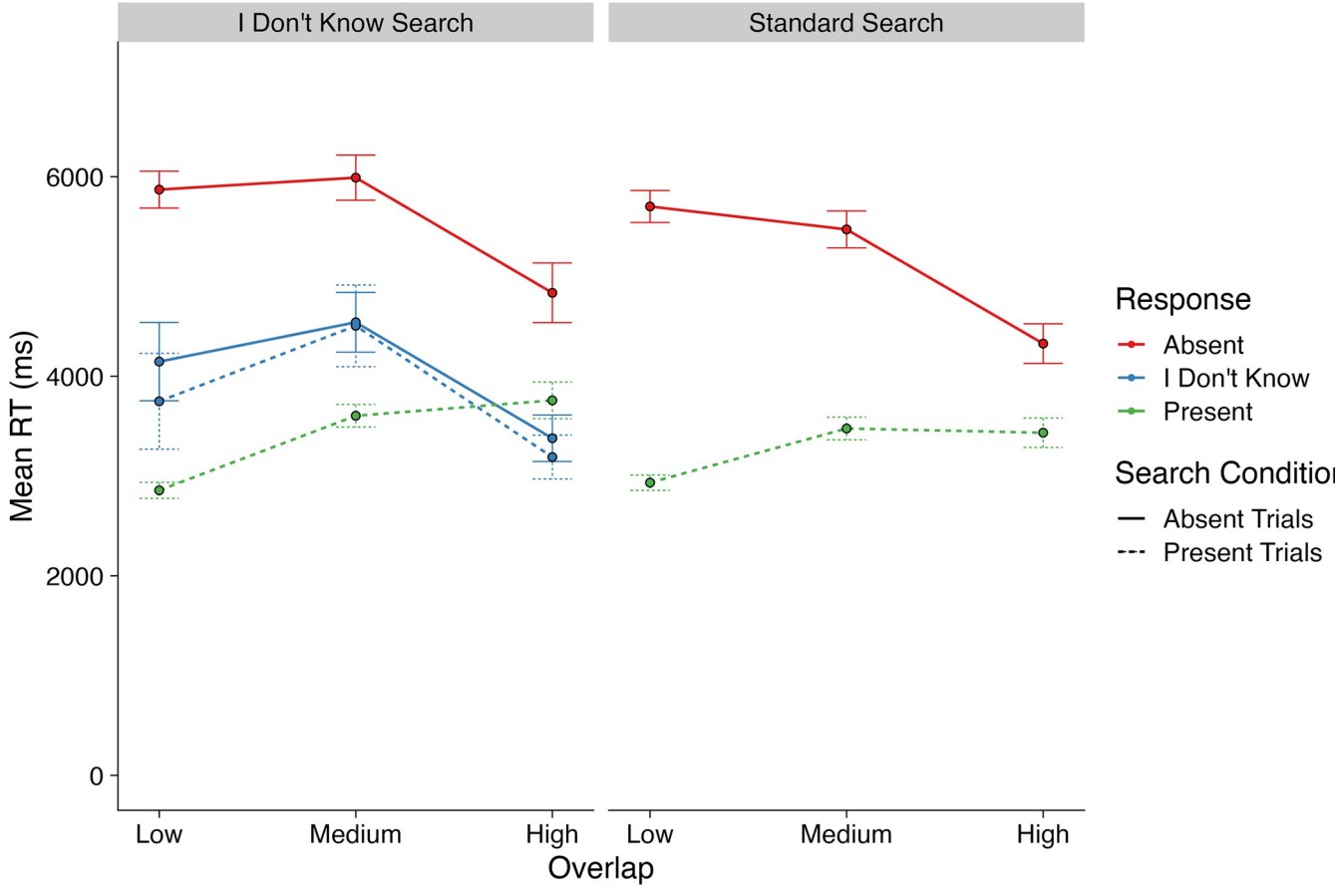

**Fig 2. Trial sequence for both search conditions.**

### Stimuli

We took our inspiration for the stimuli from previous work that has examined search of very difficult displays [17]. We then adapted this approach to more carefully control levels of overlap, in line with previous studies that have manipulated overlap levels [11].

Participants were asked to search for the lowercase letter 'o' in the displays. Distractors consisted of all of the other letters of the alphabet, also written using lowercase. The letters were written using the Arial font in size 26. We do not report these values in visual angle due to the fact that we could not measure the distance between participants and their displays in an online study.

We controlled the overlap in a similar manner to previous studies of overlap [11]. In each display, there were 20 clusters of four randomly selected letters, arranged into a virtual 5 x 5 grid. The clusters were initially centered within each grid square, and then jittered in a random distance and direction along the x and y planes by a random amount up to 20 pixels. To begin with, each of the letters was placed at the same location, meaning that they initially had a very high level of overlap.

From this high overlap starting point, the objects were 'pushed' away towards the corner of their virtual grid cell (i.e., one object was pushed towards the top-left, one towards the top-right, one towards the bottom-left, and one towards the bottom-right). Objects were pushed away further for the low level of overlap (by 15% of the cell size), less so for the medium of

overlap (by 10% of the cell size), and only by a small amount for the high level of overlap (by 5% of the cell size). After this, they were then moved towards the corner of the virtual grid cell by a random amount of up to 10 pixels in order to make the displays appear more 'random' and less ordered.

# Results

## Outliers and exclusions

We took the following steps to clean the base dataset. Since this was an online study with highly difficult and ambiguous stimuli, we took a very conservative approach to data cleaning: that is, we extensively cleaned the data to make sure that the participants retained in the final dataset were properly engaged in the task.

As noted above, we began with complete datasets from 277 participants. We began by checking for participants who, at a general level, exhibited some very fast or very slow RTs. Very fast RTs can be indicative of participants simply pressing response buttons as rapidly as they can in order to complete the study with minimal effort. Very slow RTs can be indicative of participants being distracted from the study. Indeed, the fastest RT in the study was 0.2 ms in duration, and the longest was just over 86 hours in duration (this was likely a result of the study being left 'open' in a browser tab for several days without being completed). With this in mind, we removed any participants who exhibited 5 or more RTs that were < 250 ms in duration, as well as those who exhibited 5 or more RTs that were > 30,000 ms in duration. Following these removals, we retained 206 participants (75% of the original dataset).

Having removed participants who exhibited very fast or very slow RTs, we then cleaned the data based on mean RT and accuracy levels. We removed participants who scored > 2.5 standard deviations from the mean RT or response accuracy. Following this, we retained 173 participants (62% of base dataset).

After removing trials with an RT of < 250 ms or > 20,000 ms in duration (which resulted in 1.7% of the trials being removed), the final dataset consisted of 45,951 trials from 87 participants in the Standard Search condition and 86 participants in the I Don't Know Search condition.

## Analytic approach

We conducted two overall sets of analyses: the first was focused on the responses made by participants, and the second was focused upon the RTs of those responses. We used a series of confirmatory Generalized Linear Mixed Models (GLMMs) to analyze the data. These are ideal in situations where datasets are unbalanced, as was the case here (i.e., the balance of different response types was unequal). They also offer a high level of statistical power when examining datasets because, rather than being used to examine mean accuracy or mean RTs, each trial can be entered into the analyses [18]. For accuracy data, we used binomial GLMMs; for RT data, we used GLMMs with a gamma distribution, incorporating recommendations regarding how best to analyze RTs, which can be skewed [19]. Our models began with the random structure with slopes for all effects and participants, which was then reduced in complexity when there was a failure to converge. In all cases, we present the final model that was run.

## Response rates

We began by examining how often participants made each of the different responses available to them: descriptive statistics for the rates at which different responses were made are presented in Fig 3. Because there were two responses available to participants in the Standard

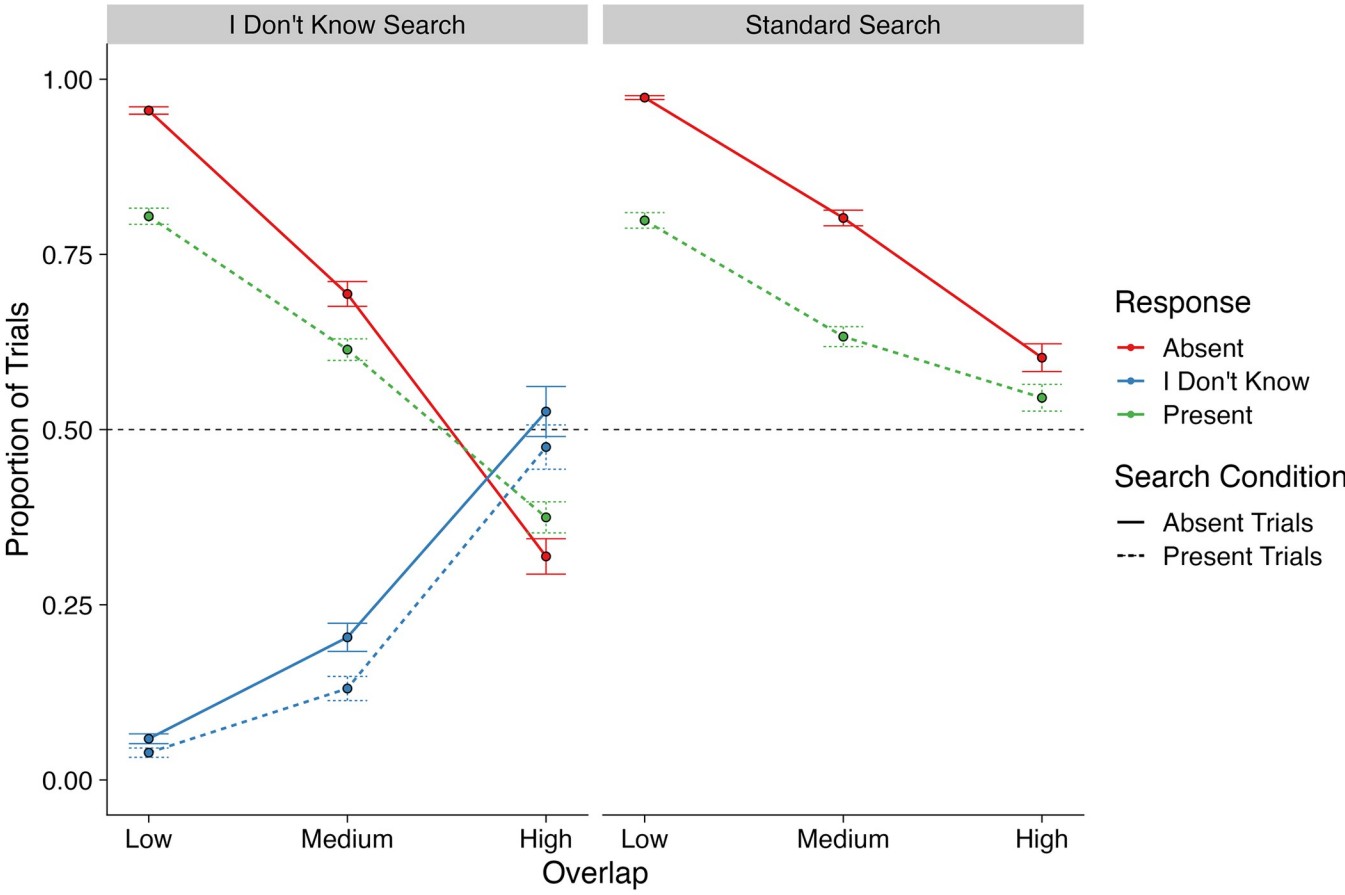

**Fig 3. Mean proportion of trials for the different responses for standard search and I don't know search.**

Search condition and three responses available to participants in the I Don't Know Search condition, this creates an imbalance in the analyses that prevents us from analyzing the entire dataset as a whole. In response to this issue, we conducted two sets of analyses when examining the response rates. Our first analysis focused on comparing 'present' and 'absent' response rates between the two Search Conditions. Our second analysis focused on the I Don't Know Search condition only and compared the rates of responding 'present' and 'absent' versus 'I don't know'.

**Response accuracy rates.** Our first GLMM examined response accuracy rates. Here, we used a binomial dependent variable where 1 = correct response and 0 = either an incorrect response *or* an 'I don't know' response for each trial. By collapsing incorrect and 'I don't know' responses together in this manner, we were able to directly compare 'present' and 'absent' response rates in the Search Conditions.

The GLMM included the following categorical factors: Search Condition (Standard Search, I Don't Know Search), Presence (Present, Absent) and Overlap (Low, Medium, High), with a full set of interactions, leading to a Search Condition x Overlap x Presence interaction. The initial GLMM included participants as a random factor, along with the full random structure for within-subjects factors. We then iterated from this random structure until a model that converged was reached: that is the model reported in Table 1.

Next, we conducted a series of contrasts to examine the interactions present within the final model. We began by examining the Search Condition x Overlap interaction. The contrasts

**Table 1. GLMM results for response rate analyses.**

| Predictors | Present/Absent | | | I Don't Know | | |
|---|---|---|---|---|---|---|
| | Log-Odds | CI | p | Log-Odds | CI | p |
| (Intercept) | 1.04 (0.04) | 0.97 – 1.12 | <**0.001** | -3.01 (0.26) | -3.52 – -2.49 | <**0.001** |
| Search Condition | 0.60 (0.08) | 0.45 – 0.75 | <**0.001** | | | |
| Overlap (Medium—Low) | -1.66 (0.05) | -1.76 – -1.56 | <**0.001** | 2.05 (0.28) | 1.50 – 2.60 | <**0.001** |
| Overlap (High—Medium) | -1.14 (0.05) | -1.25 – -1.04 | <**0.001** | 2.98 (0.16) | 2.66 – 3.30 | <**0.001** |
| Presence | -0.88 (0.07) | -1.02 – -0.73 | <**0.001** | -0.48 (0.12) | -0.72 – -0.24 | <**0.001** |
| Search Condition x Overlap (Medium—Low) | 0.12 (0.09) | -0.06 – 0.30 | 0.182 | | | |
| Search Condition x Overlap (High—Medium) | 0.91 (0.11) | 0.70 – 1.11 | <**0.001** | | | |
| Search Condition x Presence | -0.62 (0.14) | -0.90 – -0.34 | <**0.001** | | | |
| Presence x Overlap (Medium—Low) | 1.42 (0.10) | 1.22 – 1.63 | <**0.001** | 0.24 (0.22) | -0.19 – 0.67 | 0.275 |
| Presence x Overlap (High—Medium) | 0.80 (0.09) | 0.63 – 0.96 | <**0.001** | -0.31 (0.13) | -0.56 – -0.06 | **0.016** |
| Search Condition x Presence x Overlap (Medium—Low) | 0.01 (0.19) | -0.35 – 0.38 | 0.944 | | | |
| Search Condition x Presence x Overlap (High—Medium) | -0.32 (0.17) | -0.66 – 0.01 | 0.057 | | | |

confirmed our prediction that, for both Search Conditions, response accuracy reduced as Overlap increased (all $zs > 7.1$, $ps < .0001$). In addition, response accuracy was higher for Medium and High Overlap trials in Standard Search than I Don't Know Search ($zs > 3.4$, $ps < .001$), but not for the Low Overlap trials ($z < 1$, $p > .5$).

Turning to the Condition x Presence interaction, our contrasts revealed that response accuracy was higher for target-absent trials than target-present trials in both Search Conditions ($zs > 6.4$, $ps < .0001$). Alongside this, response accuracy was higher for both target-present and target-absent trials in Standard Search than I Don't Know Search ($zs > 2.3$, $ps < .05$).

Finally, we examined the Overlap x Presence interaction. Here, again as with our previous contrasts, it was clear that, for both target-present and target-absent trials, increases in Overlap resulted in decreases in response accuracy ($zs > 8.1$, $ps < .0001$). For Low and Medium Overlap trials, response accuracy was higher for target-absent than target-present trials ($zs > 6.4$, $ps < .0001$). However, this was not the case for High Overlap trials ($z = 1.7$, $p = .09$).

Overall, our initial examination of response accuracy rates revealed that increasing overlap had a deleterious effect on response accuracy, which was expected. Moreover, it demonstrates that increasing levels of overlap was effective in impairing search performance, almost to chance level for High Overlap trials. For some experiments this would prove to be problematic, but this was an intended feature of the present study. Most importantly, as we had predicted, response accuracy rates were higher for Standard Search than I Don't Know Search: the next analysis that we conducted examined whether this shift in responding was due to a rise in the rate of 'I don't know' responses from participants.

**'I don't know' response rates.** Next, we examined the rate at which participants responded 'I don't know' (in the I Don't Know Search condition only, of course). This

involved conducting a second binomial GLMM that once again coded 1 = correct response, but here 0 = 'I don't' know' responses. We excluded trials in which an incorrect response had been given to mirror the previous set of analyses. The model included the same Presence and Overlap factors as the previous GLMM. For target-present trials, because we included only 'present' or 'I don't know' responses in this analysis, the results here provided us with a comparison between the rate of 'present' versus 'I don't know' responses. Likewise, for target-absent trials, because we included only 'absent' or 'I don't know' responses, the results for target-absent trials compared the rate of 'absent' versus 'I don't know' responses. The results for this model are presented in Table 1.

Because the model included a Presence x Overlap interaction, we conducted a series of contrasts to examine this interaction in detail. The contrasts revealed that, as expected, as Overlap increased, the rate of 'I Don't Know' responses increased for both target-present and target-absent trials, with the largest increase in 'I Don't Know' response rates being between the Medium and High Levels of Overlap (all $zs > 5.4$, all $ps < .0001$). In addition, there was no difference in the rate of 'I Don't Know' responses between target-present and target-absent trials for any level of Overlap ($zs < 1$, $ps > .44$).

Our analyses of the rate of 'I don't know' responses complement the response accuracy analyses by demonstrating that, as expected, when Overlap increased in the I Don't Know Search condition, although response accuracy reduced, the rate at which participants responded 'I don't know' increased. Again, this suggests that the experimental manipulation of increasing overlap was successful in making the displays highly ambiguous to examine. Moreover, it also suggests that the higher response accuracy that we found in the Standard Search condition could have been the result of a high rate of guessing from the participants: faced with no other choice than 'present' or 'absent', participants may have simply been forced to make a guess for the higher levels of overlap in the Standard Search condition.

## Response times

Next, we examined Response Times. We took a similar approach here to our examination of response rates described above. We used two GLMMs to compare RTs for target-present and target-absent trials across both Search Conditions (first GLMM), and then to compare RTs for 'present', 'absent' and 'I don't know' responses in the I Don't Know Search condition only. Descriptive statistics are presented in Fig 4 and the GLMM models used for the analyses of RTs are presented in Table 2.

**'Present' and 'absent' response RTs.**   We began by examining RTs in correct-response trials, excluding the 'I don't know' responses from the I Don't Know Search condition. The GLMM that we conducted included the following factors: Search Condition (Standard Search, I Don't Know Search), Presence (Present, Absent) and Overlap (Low, Medium, High), with a full set of interactions, leading to a Search Condition x Overlap x Presence interaction. The initial GLMM included participants as a random factor, along with the full random structure. We then iterated from this random structure until a model that converged was reached: this final model is presented in Table 2.

Given the significant Search Condition x Presence x Overlap interaction, we then conducted a series of contrasts to understand the source of the interaction. These contrasts revealed that target-absent RTs were longer than target-present RTs in all Overlap levels and Search Conditions ($ts > 642$, $ps < .0001$). They also revealed that, for target-absent trials, RTs decreased as Overlap increased ($ts > 59$, $ps < .0001$), whilst for target-present trials, RTs increased as Overlap increased ($ts > 25$, $ps < .0001$). Examining Fig 4, it is quite difficult to see this effect at the level of the means. However, it can be seen more clearly in Fig 5, which plots

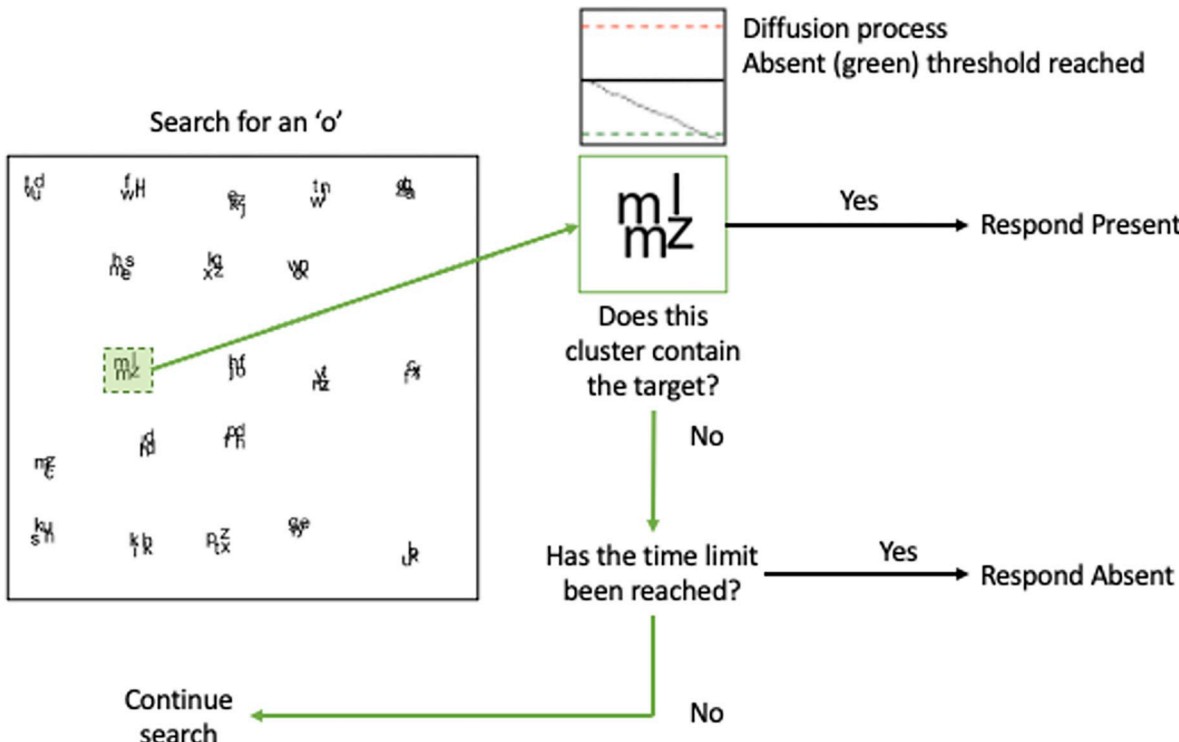

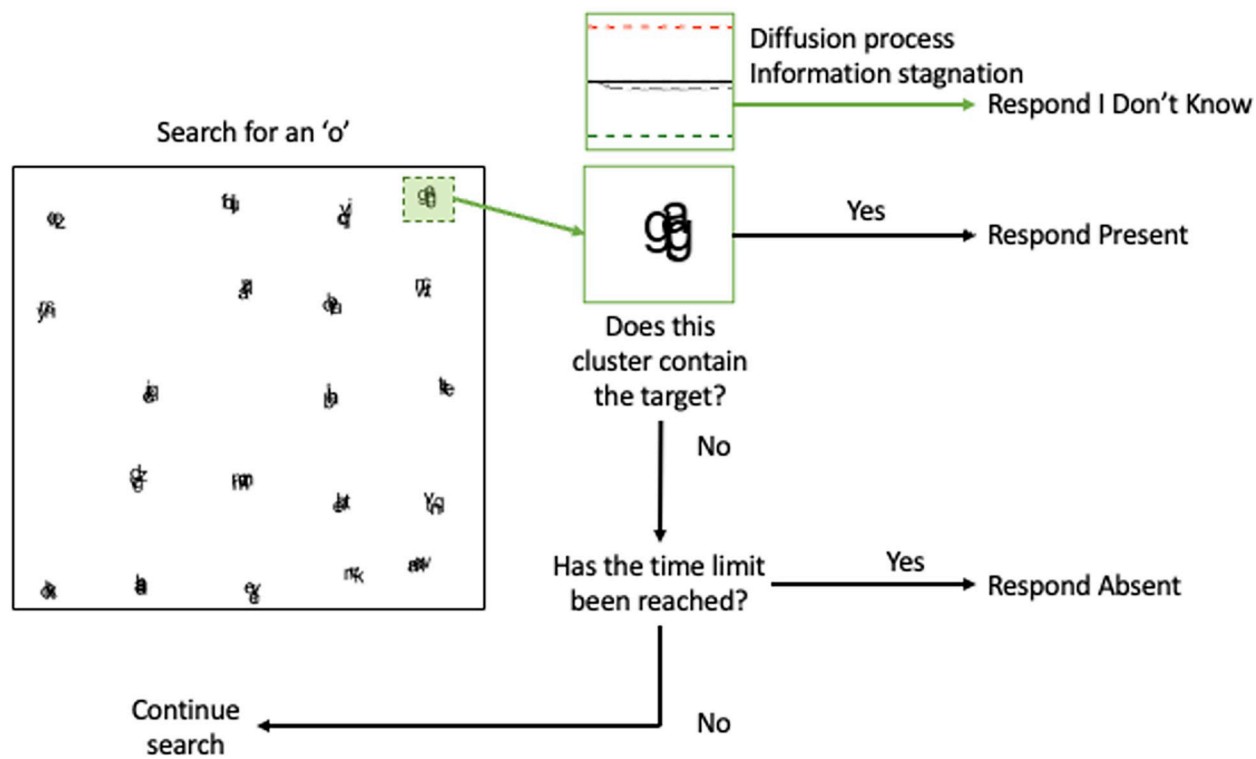

**Fig 4. Mean response times for the different responses for standard search and I don't know search.**

**Table 2. GLMM results for response time analyses.**

| Predictors | Present/Absent | | | I Don't Know | | |
|---|---|---|---|---|---|---|
| | Estimates | CI | p | Estimates | CI | p |
| (Intercept) | 4401.20 (1.32) | 4398.62 – 4403.79 | <**0.001** | 3805.83 (0.00) | 3805.83 – 3805.84 | <**0.001** |
| Search Condition | -287.27 (1.28) | -289.77 – -284.76 | <**0.001** | | | |
| Overlap (Medium—Low) | 101.27 (1.12) | 99.07 – 103.47 | <**0.001** | -944.49 (0.00) | -944.49 – -944.49 | <**0.001** |
| Overlap (High—Medium) | -555.70 (1.90) | -559.43 – -551.97 | <**0.001** | | | |
| Presence | -2097.34 (1.06) | -2099.41 – -2095.26 | <**0.001** | -1032.45 (0.00) | -1032.45 – -1032.45 | <**0.001** |
| Search Condition x Overlap (Medium—Low) | -217.21 (1.08) | -219.34 – -215.09 | <**0.001** | | | |
| Search Condition x Overlap (High—Medium) | -180.99 (1.04) | -183.04 – -178.95 | <**0.001** | | | |
| Search Condition x Presence | 298.49 (1.13) | 296.27 – 300.71 | <**0.001** | | | |
| Presence x Overlap (Medium—Low) | 800.46 (1.22) | 798.08 – 802.85 | <**0.001** | 783.77 (2.14) | 779.57 – 787.96 | <**0.001** |
| Presence x Overlap (High—Medium) | 981.30 (1.36) | 978.64 – 983.97 | <**0.001** | | | |
| Search Condition x Presence x Overlap (Medium—Low) | 33.50 (1.00) | 31.54 – 35.46 | <**0.001** | | | |
| Search Condition x Presence x Overlap (High—Medium) | -141.35 (1.50) | -144.29 – -138.40 | <**0.001** | | | |
| Response | | | | -805.09 (0.00) | -805.10 – -805.09 | <**0.001** |
| Response x Overlap (High—Medium) | | | | -501.07 (0.00) | -501.07 – -501.06 | <**0.001** |
| Presence x Response | | | | 861.67 (0.00) | 861.67 – 861.68 | <**0.001** |
| Presence x Response x Overlap (High—Medium) | | | | -432.16 (0.00) | -432.17 – -432.16 | <**0.001** |

the overall RT distributions for the different conditions and levels of Overlap. For example, for target-absent trials in the I Don't Know search condition, the mean RT increased between Low and Medium Overlap, but for our model, and at the distributional level, a clear shift towards faster RTs can be seen. Finally, we also found that RTs in the I Don't Know Search condition were longer than those in the Standard Search condition for all levels of Overlap and Presence ($ts > 97$, $ps < .0001$).

Overall, results for the RT data were not entirely in line with our predictions. We expected that 'present' RTs would increase as Overlap increased, and this was confirmed to be the case. We expected the same for target-absent RTs, but instead found the opposite to be true. Still, RTs for 'absent' responses were slower than for 'present' responses in both Search Conditions, as expected. The remaining issue to be addressed is how RTs for 'I don't know' responses compared to 'present' and 'absent' response RTs.

**RTs during 'I don't know' search.** Finally, we examined RTs focusing on I Don't Know Search only. The GLMM that we used here included the Overlap and Presence factors, as in our previous GLMMs, but also introduced a new factor: Response, and it indicated the response given by participants on each trial. This was a categorical factor with two levels:

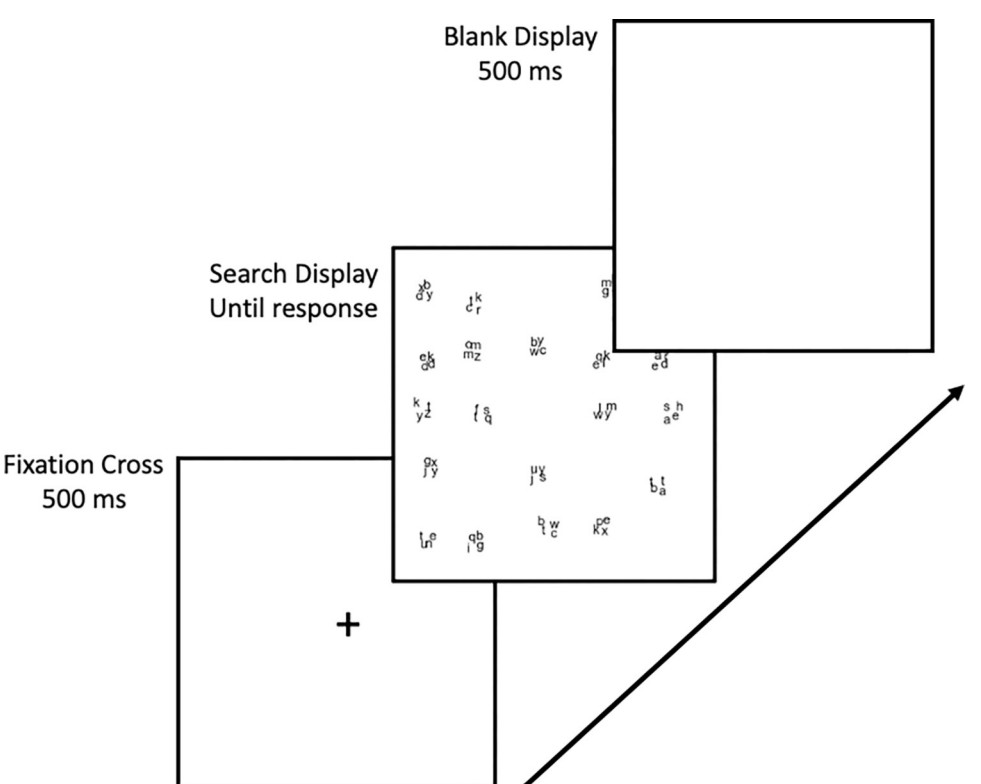

**Fig 5. Density plots showing RT distributions for the search conditions at different levels of overlap.**

namely 'I don't know' and 'correct'. Since we excluded incorrect response trials from our analyses, the results here let us compare RTs for 'I don't know' responses versus 'absent' responses (by looking at the effects of Response in target-absent trials), as well as letting us compare RTs for 'I don't know' versus 'present' responses (by looking at the effects of Response in target-present trials). We excluded Low Overlap trials from this analysis because participants rarely responded 'I don't know' on these trials (they did so for only 124 trials in total), and as such, this did not permit a sufficiently powered or meaningful analysis of RTs for these trials As with the other GLMMs, we began with the full random structure and then iterated down until a converging model was reached. The final model is presented in Table 2.

Because the final model included a significant Overlap x Presence x Response interaction, we then sought to examine this interaction in detail with a series of contrasts. These revealed that, as Overlap increased, RTs for 'I don't know' responses decreased, as was expected ($ts > 430.4$, $ps < .0001$). They also revealed that participants were faster to respond 'I Don't Know' on target-absent trials than they were to respond 'absent' on target-absent trials ($ts > 269.2$, $ps < .0001$). For target-present trials with Medium Overlap, 'I don't know' RTs were slower than 'present' RTs ($t = 40.52$, $p < .0001$), yet for target-present trials with High Overlap, this pattern was reversed, with 'I don't know' RTs being *faster* than 'present' RTs ($t = 87.66$, $p < .0001$).

Our analyses of the 'I don't know' RTs helps to complete the picture regarding search and decision-making in our study. 'I don't know' responses were more rapid than 'absent' RTs and slower than 'present' RTs, but only when overlap was relatively low. When overlap was high, RTs for 'I don't know' responses were more rapid than for 'present' responses. Though a rather complex pattern of results, these were in line with our predictions.

## Discussion

Standard visual search experiments, as well as the models and theories built upon those experiments, focus on search tasks that involve giving participants two response options: 'present' or 'absent'. This stands in contrast to many real-world search tasks that involve a range of other options that are available to a searcher, including radiography, airport screening and other tasks such as information searches. These real-world search tasks often include the ability to respond with some degree of indecision when the task cannot be completed with certainty. Here, we focused on one such response option: 'I don't know'. In order to understand how participants respond when given the ability to indicate 'I don't know' in a visual search task, we asked participants to search through displays that contained varying levels of object overlap. This introduced a level of ambiguity in resolving individual clusters that led to participants being unable to definitively answer 'present' or 'absent' in some circumstances. We reasoned that the higher levels of overlap would be so difficult to examine that it would be effectively impossible for participants to determine the correct answer. As a result, this would encourage them to give an 'I don't know' response, should they wish to do so.

Our manipulation of overlap levels was a clear success: response accuracy dropped for both target-present and target-absent trials as overlap increased. For the high overlap trials, response accuracy was at chance (for the Standard Search condition, mean accuracy in high overlap was 0.5, $sd$ = 0.09). If this situation had arisen for a standard visual search task, chance levels of accuracy would of course be problematic. Here, however, it was a key part of the study in ensuring that we were using highly difficult and ambiguous stimuli with no clear, easily discernable answer.

When designing the study, our predictions were based upon an extension of the Guided Search model of visual search [2]. We reasoned that, when faced with a cluster of objects that were too difficult to resolve and determine whether a target was present or not, the object identification process would suffer from what we have termed *information stagnation*. That is, beyond a certain point, a diffusion process that travels towards a 'present' or 'absent' decision for a cluster might halt making any progress towards either of its two decision thresholds. As shown in Fig 1, we expected that once information stagnation occurred, an 'I don't know' response would be generated by participants (if they were given that option). When faced with only a 'present' or an 'absent' response, we expected that participants would simply guess (under these circumstances). This, we expected, would reduce response accuracy for 'I don't know' search because, rather than make a guess and have a 50% chance of being correct, participants would instead respond 'I don't know'.

Our results were in line with these predictions. Response accuracy was higher for the standard search condition compared with the 'I don't know' search condition because, as overlap increased, participants in the 'I don't know' search condition opted to respond 'I don't know' rather than being forced to make a 'present' or 'absent' response. Analysis of the RT data did not, however, entirely align with our predictions. For target-present trials, RTs increased as overlap increased, and we expected this to occur based on prior, similar research [11]. For target-absent trials, however, RTs decreased as overlap increased, contrary to our expectations and to prior research. We believe that, as overlap increased, for each cluster that participants examined, the probability that any one cluster would generate an 'I don't know' decision increased. If that was the case, then the reduced RTs (as overlap increased) would be due to participants rapidly reaching such a cluster and responding 'I don't know' (when they were able to), or simply guessing (in Standard Search).

Overall, we found that 'I don't know' RTs were more rapid than 'absent' responses. We had expected this given that, when a cluster that generated an 'I don't know' response was

examined, this would likely occur before a complete and exhaustive search of the display that would lead to an 'absent' response. 'I don't know' RTs were slower than 'present' RTs for the low and medium overlap trials only, yet were faster than 'present' RTs for high overlap trials. This again we had expected because, when the number of clusters in a display that generated an 'I don't know' response was quite small (as is the case in low and medium overlap trials), the target would often be found before one of these clusters was first encountered. However, when the number of clusters in a display that generated an 'I don't know' response was very high (as was the case in the high overlap trials), then perhaps even the first or second cluster could trigger an 'I don't know' response, hence the RTs for 'I don't know' responses become faster even than 'present' responses.

Until now, visual search tasks have largely been framed in terms of binary present/absent and correct/incorrect outcomes. This enables a high level of scientific control over experiments, tasks, and stimuli in the laboratory, but is quite removed from the reality of real-world searches where there are many possible outcomes, including even the ability to admit that one simply does not know the answer. Our results raise questions about the findings of other previous studies that have used highly complex search stimuli, including both studies with overlapping stimuli, as well as studies with stimuli that are ambiguous, complex or difficult to detect [1,11,20]. By not giving participants the ability to respond 'I don't know' in these previous studies, it may be the case that much of what appears to be shifts in response accuracy is simply a case of participants making guesses under certain conditions. Future research could therefore, we feel, benefit from including 'I don't know' responses when ambiguous or highly difficult stimulus sets are used.

Our study has important implications for models and theories of visual search. As noted in the Introduction, we focused on the prominent Guided Search model as a starting point for our conceptualization of how 'I don't know' responses might be generated. We chose this model because it shared a common structure with most other models of search [3–8]. As such, our findings demonstrate that these models have a clear limitation in the sense that they are not able to capture 'I don't know' responses from searchers when faced with ambiguous stimuli. Our information stagnation account offers one simple and parsimonious approach to updating those models in order to resolve this issue–and these models may well benefit from being modified to include information stagnation in their structure.

The model and account regarding information stagnation that led to 'I don't know' responses are, as we noted, just one of many possible ways of capturing participants' decisions and responses to highly ambiguous stimuli. As we noted in the Introduction, there are a number of different routes by which an 'I don't know' response might be generated: perhaps participants responded 'I don't know' after examining one cluster of objects for which they could not reach a 'Present' or 'Absent' decision; perhaps the 'I don't know' responses were generated after *multiple* clusters are examined for which a decision could not be reached. Another alternative is that the 'I don't know' responses were generated based upon an overall impression of how difficult a display was to examine.

We focused on the first of these options here for our account given that it is the simplest possible account, and that seemed to be a sensible starting point for this first examination of 'I don't know' responses in visual search. That being said, future research would benefit from teasing apart the mechanics of how these 'I don't know' responses are generated. This could be done in a variety of different ways. Future studies could, for example, ask participants to provide a confidence rating of their responses, which would tap into the possibility that participants were simply not particularly confident about their responses when they responded 'I don't know', rather than not knowing the answer at all. Another possibility is that future research could adopt a two-stage response approach wherein participants first make a response

to a trial in the same manner as the present study, and then click on a cluster that they believe might contain the target. This approach could aid in distinguishing between scenarios where participants do not know whether a target is present *anywhere* in a display versus scenarios where participants do not know that a specific item or cluster contains a target (or even, might contain a target). Overall, embracing the grey area between 'present' and 'absent' responses, we believe, will be highly valuable in further developing our understanding of responses during visual search.

We also believe that future research could take the methodology developed here further to help better understand effects wherein searches are biased towards one response over another. This could involve, for example, presenting targets on a small proportion of trials. Known as the effect of target 'prevalence' [21], it has been reported extensively that, when targets only appear rarely, searchers become biased towards 'absent' responses. It would of interest to determine whether participants respond 'I don't know' at the same rate regardless of target prevalence when faced with increases in overlap in the stimuli. This could be tested at both low (<50%) prevalence levels and high (>50%) prevalence levels. In these cases of low or high prevalence, searchers may be sufficiently biased towards or away from target detection that they become less willing to respond 'I don't know' and instead respond 'present' often when prevalence is high and overlap is high and respond 'absent' often when prevalence is low and overlap is high.

Beyond this, embracing the ambiguity that is inherent to the task that we have used here could be valuable in further understanding the effects of a variety of individual differences in searchers. One particularly interesting option in this regard would be to use this approach to better understand *intolerance of uncertainty*. Intolerance of uncertainty is a key factor in a range of anxiety disorders, and, using simple search tasks, it has been found that intolerance of uncertainty has a clear impact upon search performance and behavior [22–26]. However, it remains unclear what effect intolerance of uncertainty will have when participants are faced with highly difficult overlapping stimuli like those that were used here, and also given the opportunity to respond 'I don't know'.

## Acknowledgments

The authors would like to thank Jeremy Wolfe and two anonymous reviewers for their comments on an earlier version of this manuscript.

## Author Contributions

**Conceptualization:** Hayward J. Godwin, Michael C. Hout.

**Data curation:** Hayward J. Godwin.

**Formal analysis:** Hayward J. Godwin.

**Investigation:** Hayward J. Godwin.

**Methodology:** Hayward J. Godwin, Michael C. Hout.

**Project administration:** Hayward J. Godwin.

**Resources:** Hayward J. Godwin.

**Software:** Hayward J. Godwin.

**Validation:** Hayward J. Godwin.

**Visualization:** Hayward J. Godwin, Michael C. Hout.

Writing – **original draft:** Hayward J. Godwin.

Writing – **review & editing:** Hayward J. Godwin, Michael C. Hout.

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
