## [Decision Letter · Decision Letter 0]

2 Oct 2023

PONE-D-23-15409Just Say ‘I Don’t Know’: Understanding Information Stagnation during a Highly Ambiguous Visual Search TaskPLOS ONE

Dear Dr. Godwin,

Thank you for submitting your manuscript to PLOS ONE. After careful consideration, we feel that it has merit but does not fully meet PLOS ONE’s publication criteria as it currently stands. Therefore, we invite you to submit a revised version of the manuscript that addresses the points raised during the review process.

We look forward to receiving your revised manuscript.

Kind regards,

Sathishkumar Veerappampalayam Easwaramoorthy

Academic Editor

PLOS ONE

Journal Requirements:

Reviewers' comments:

Reviewer's Responses to Questions

**Comments to the Author**

1. Is the manuscript technically sound, and do the data support the conclusions?

Reviewer #1: Yes

Reviewer #2: Yes

2. Has the statistical analysis been performed appropriately and rigorously? 

Reviewer #1: Yes

Reviewer #2: Yes

3. Have the authors made all data underlying the findings in their manuscript fully available?

Reviewer #1: Yes

Reviewer #2: Yes

4. Is the manuscript presented in an intelligible fashion and written in standard English?

Reviewer #1: Yes

Reviewer #2: Yes

5. Review Comments to the Author

Reviewer #1: This article presents a "simple" study in the best sense of the word. The addition of a "I don't know" response condition to visual search situations is innovative. The study itself is well conducted, with appropriate and clear methodology and analyses. The manuscript is very clear and easy to read.

Nevertheless, I would have liked to see a more in-depth discussion of the theoretical implications of the results. Of course, we understand what the authors are stressing about the importance of adding the possibility of an "I don't know" response in future studies, but to what results would this inclusion lead to, and how can such results lead to at least a partial revision of current theoretical models of visual search?

I guess this issue would be easier to address in the discussion if more theoretical elements were presented in the introduction (here only one model is presented in the introduction and few results from previous studies in the field). I'm not a specialist of visual search, but this seems a bit reductive.

Without weighing down the paper, which is very clearly written, I would nevertheless encourage the authors to explain and clarify the theoretical implications of their results and of their innovation in terms of methodology.

In terms of 'practical' implications, and beyond the anecdotal examples mentioned in the introduction, I think that these results can also be highly relevant in more complex search situations such as information search within texts or documents, where the presence/absence of the target (e.g., answer to a question) is generally not obvious. Perhaps the authors could at least mention such possible implications in the discussion?

Finally, the prevalence of 'I don't know' responses is only discussed in terms of contextual factors (such as the presence of ambiguous situations with, for example, the high overlap display). But could it also be possible that the ability to respond 'I don't know' is influenced by certain individual characteristics (such as age, level of commitment to the task or other (meta)cognitive factors)?

Reviewer #2: This is a nice, well conducted, thoroughly analysed study. The findings will be interested to many readers of the journal, especially those in cognitive psychology and related fields. I have no major concerns that would prevent me from recommending publication- it’s a neat paper and a nice contribution.

I applaud the authors for providing all materials, data, and analyses. However, some instructions for running the experiment would be useful. I also note that the trimmed data set was included. It would be good to provide the full data set, or at some indication of what the data looked like when culled at 250 ms and 30,000 ms- did the same pattern of data emerge? The approach (I think rightly) that the authors took was conservative, so I’d just like to be reassured that a more liberal approach didn’t result in a very different pattern of data. However, I would encourage the authors to also consider pre-registration in the future too (this would particular benefit my previous point about culling data, pre-registering this is much safer and more transparent).

I would like slightly more space devoted to the concept of what participants understand by an ‘I don’t know’ response. The exact instructions to participants would be useful here (I’m afraid I didn’t have time to get the experiment to run, but even if I had, I think the casual reader would appreciate this upfront in the article since it is the key novel addition of this study). As the authors rightly pick-up in their discussion, the response could be used in several ways both between and within participants. For example, it may be used when they run out of time and have yet to decide, it may be used as soon as they find an area that is ambiguous, and they are not sure whether it contain a target. It may also be used as a pure guess or error on some trials. Further investigating these possibilities will help address why the RT pattern did not follow predictions or previous findings. Future work may consider looking at a two-stage response where participant provide a confidence rating or are made to ‘guess’ after an ‘I don’t know’ response (the recognition memory literature is potentially useful here for more nuanced response scales). Another method could be to have participants subsequently click where they guess (by selecting the location) the target might be (or to use eye tracking and link gaze location at time of response). I certainly not suggesting these need to be run here, but some indication of future work might be useful.

6. PLOS authors have the option to publish the peer review history of their article (what does this mean?). If published, this will include your full peer review and any attached files.

Reviewer #1: No

Reviewer #2: No

---

## [Author Response · Author response to Decision Letter 0]

9 Nov 2023

Response Letter

Dear Dr Sathishkumar,

Thank you for the consideration of our manuscript for publication in PLOS ONE. We very much appreciate the level of detail and insightful comments provided by yourself and the reviewers. We were pleased to see that the reviewers were, on the whole, very positive about the manuscript indeed. We have taken the time to carefully consider the points raised in response to the first submission of the manuscript, as outlined in this letter. As a result, we have made a number of changes to the manuscript.

We have also included responses to a review from a third reviewer. This is somewhat unusual and we recognise that fact. The third reviewer emailed the first author (HG) their review stating that their reviews have, on occasion, not gone through properly in the PLOS ONE system. In the interests of transparency and fairness to the time and effort put in by this reviewer, we have responded to the comments from this reviewer as well.

We have highlighted the relevant changes within the manuscript itself with track changes as requested.

Sincerely,

Hayward Godwin and Michael Hout

Reviewer #1

This article presents a “simple” study in the best sense of the word. The addition of a “I don’t know” response condition to visual search situations is innovative. The study itself is well conducted, with appropriate and clear methodology and analyses. The manuscript is very clear and easy to read.

We would very much like to thank this reviewer for their positive comments here.

Nevertheless, I would have liked to see a more in-depth discussion of the theoretical implications of the results. Of course, we understand what the authors are stressing about the importance of adding the possibility of an “I don’t know” response in future studies, but to what results would this inclusion lead to, and how can such results lead to at least a partial revision of current theoretical models of visual search?

I guess this issue would be easier to address in the discussion if more theoretical elements were presented in the introduction (here only one model is presented in the introduction and few results from previous studies in the field). I’m not a specialist of visual search, but this seems a bit reductive.

Without weighing down the paper, which is very clearly written, I would nevertheless encourage the authors to explain and clarify the theoretical implications of their results and of their innovation in terms of methodology.

We thank the reviewer for raising this issue. We have now expanded upon why we have used the Guided Search model as the starting point in the Introduction – specifically, because its basic structure is common to virtually all models of visual search (see page 5).

We have also been much more specific about revisions to this basic structure of models of search in the Discussion section, explicating what such changes need to look like (see page 33 onwards).

In terms of ‘practical’ implications, and beyond the anecdotal examples mentioned in the introduction, I think that these results can also be highly relevant in more complex search situations such as information search within texts or documents, where the presence/absence of the target (e.g., answer to a question) is generally not obvious. Perhaps the authors could at least mention such possible implications in the discussion?

We would like to thank this reviewer for highlighting this to us, we have included a note regarding information searches in the Introduction (page 3), and the Discussion (page 30). 

Finally, the prevalence of ‘I don’t know’ responses is only discussed in terms of contextual factors (such as the presence of ambiguous situations with, for example, the high overlap display). But could it also be possible that the ability to respond ‘I don’t know’ is influenced by certain individual characteristics (such as age, level of commitment to the task or other (meta)cognitive factors)?

Thanks to this reviewer for this suggestion. We have now included some discussion around individual differences, focusing on one specific individual difference that we believe would be particularly interesting for further study. Namely, this involves intolerance of uncertainty (page 36)

Reviewer #2

This is a nice, well conducted, thoroughly analysed study. The findings will be interested to many readers of the journal, especially those in cognitive psychology and related fields. I have no major concerns that would prevent me from recommending publication- it’s a neat paper and a nice contribution.

We would very much like to thank this reviewer for their positive comments here.

I applaud the authors for providing all materials, data, and analyses. However, some instructions for running the experiment would be useful. I also note that the trimmed data set was included. It would be good to provide the full data set, or at some indication of what the data looked like when culled at 250 ms and 30,000 ms- did the same pattern of data emerge? The approach (I think rightly) that the authors took was conservative, so I’d just like to be reassured that a more liberal approach didn’t result in a very different pattern of data. However, I would encourage the authors to also consider pre-registration in the future too (this would particular benefit my previous point about culling data, pre-registering this is much safer and more transparent).

Thanks to this reviewer for their comments here. We have now included a guide with regards to how to get the experiment running as part of the online shared materials.

We have also included in the online shared materials the raw data alongside the code used to clean it to a point that it was ready for analyses.

We have taken a cursory examination of re-running the analyses with the higher limit of RTs as suggested here (i.e., going from 15,000 ms as in the manuscript to the 30,000 ms). For the first set of RT analyses, this resulted in a situation wherein the model that we ran would no longer converge with the larger dataset included. As such, this poses a problem because the solution to a model that does not converge initially is to reduce the random structure of the model. Reducing the random structure of the model does, unfortunately, result in the model being less conservative. Ultimately this means that here having a less conservative data filter would result in necessitating a less statistically conservative model. For that reason, we have not proceeded with such cross-comparisons in detail. Moreover, we do stand by our approach to be highly conservative with our data filtering since this was an online study where we simply had no idea and no way to check whether participants were truly in a distraction free environment or not. 

I would like slightly more space devoted to the concept of what participants understand by an ‘I don’t know’ response. The exact instructions to participants would be useful here (I’m afraid I didn’t have time to get the experiment to run, but even if I had, I think the casual reader would appreciate this upfront in the article since it is the key novel addition of this study). As the authors rightly pick-up in their discussion, the response could be used in several ways both between and within participants. For example, it may be used when they run out of time and have yet to decide, it may be used as soon as they find an area that is ambiguous, and they are not sure whether it contain a target. It may also be used as a pure guess or error on some trials. Further investigating these possibilities will help address why the RT pattern did not follow predictions or previous findings. Future work may consider looking at a two-stage response where participant provide a confidence rating or are made to ‘guess’ after an ‘I don’t know’ response (the recognition memory literature is potentially useful here for more nuanced response scales). Another method could be to have participants subsequently click where they guess (by selecting the location) the target might be (or to use eye tracking and link gaze location at time of response). I certainly not suggesting these need to be run here, but some indication of future work might be useful.

Thank you for these helpful suggestions. We have now included detailed information and the rationale behind the instructions given to participants in the Method section (pages 11 and 12). 

We agree entirely about the two-stage response approach and what participants understand by an ‘I don’t know’ response, which was touched upon by Reviewer #3 as well. We believe that it would be a very valuable avenue for future development to better pick apart the ‘I don’t know’ responses in detail. We have now included a description of this in the Discussion section (pages 34 and 35). 

Reviewer #3

I really like the basic idea here. The standard 2AFC yes/no visual search paradigm does not allow for the possibility that you just aren’t sure about the answer and might like the option to answer “I don’t know”. The experiment compares the 2AFC paradigm with a 3AFC paradigm that allows “I don’t know”. The experiment finds some fairly obvious things. E.g. the rate of I don’t know goes up as the task gets harder. I don’t find the overall package to be as satisfying as I had hoped. What are my concerns?

2) Starting with the set-up of the problem. I think that there are two kinds of “I don’t know”. There is the situation where I don’t know IF the there is a target here at all. That is not quite the same as “I don’t know if THIS item is a target”.

We’d like to thank Reviewer #3 for pointing this out. We have now included a more detailed and nuanced section on these and related issues in the Discussion (pages 34 and 35). 

2) Your data cleaning seems a bit extreme. You are losing a lot of Os. P14 you say, “ With this in mind, we removed any participants who exhibited 5 or more RTs that were < 250 ms in duration, as well as those who exhibited 5 or more RTs that were > 30,000 ms in duration.” I can see that you would want to filter out RTs that are too low or too high, but why toss the whole subject for just five bad responses? 

As noted in the response to Rev2 above, we are confident that the conservative data cleaning approach is appropriate here. This is because, since this is an online study, we have few avenues by which to be certain that participants are engaging in the task appropriately. Removing participants with even a few long RTs has become a common practice in our labs after finding that some participants will, for example (we assume) leave their computers for many minutes or even hours before returning to them. This stringent filter enables us to be certain that the datasets which are retained are from participants who engaged in one continuous run of the study with as few distractions or breaks from the task as possible. 

3) I find the data figures confusing. I think I would plot the 2AFC and 3AFC data on separate panels. 

4) Fig 3: Why don't Absent and I don't know responses add to 100%? Oh, I suppose because there are FA erors in there. Why don't you plot the stadard error data on the same graphs? So you would have two lines on each of the 2AFC graphs and 3 lines on each 3AFC graph.

We thank the reviewer for raising these suggestions – we have now replotted Figures 3 and 4 in line with what has been suggested here (page 17 and page 24).

5) So, let's look at Medium Present trials. Looks like there would be about 35% Miss errors in the 2AFC condition. Those 35% would be divided roughly equally between "no" (Miss) responses and "I don't know" responses in the 3AFC case. This makes me wonder if, in this particular experiment, the "I don't know" responses are just low-threshold "no" responses (or maybe it would be better to say low confidence). On the absent trials, these low threhsold "no" responses cut down on the proportion of correct "no" responses.

This is an interesting point, and we’d like to thank the reviewer for it. In line with the suggestions raised by Reviewer #2, we’ve added some discussion around confidence and other types of ‘I don’t know’ response to the Discussion (see pages 34 and 35).

6) Fig 3 on p24 is really Fig 4.

Thanks to this reviewer for catching this error – we have updated the Figure number accordingly (page 24).

7) This single experiment makes for a pretty thin paper. There are lots of stats but they aren't telling us a whole lot that isn't rather self-evident. There are lots of things you could try, since these are easy experiments. Different stimuli (e.g. stimuli that are hard but not visually ambiguous….high set size color X color conjunctions might work). You mention prevalence. That could be interesting. Changing the reward structure so there is a cost for "I don't know". Etc….but something more than just this one experiment. 

Though this does indeed involve the presentation of a single study, the sample size is quite large for a “typical" visual search experiment. Moreover, we understand that this particular journal is more focused on publishing empirical research regardless of the number of experiments or the degree to which the findings are substantive. That being said, we do agree that there are many potential further avenues that could be explored here. Frankly, we were keen to go through the peer review process after completing this study before conducting any follow-ups to gain feedback on the design and make any changes before launching into a broader programme of research in this area. Indeed, both of our labs are currently in the process of starting more research into these possibilities.

8) I do like the emphasis on Guided Search….but then I am a bit biased! Thanks.

We thought this might be the case!

In sum, cool idea. Well worth exploring. The current story doesn't explore enough for my taste.

Best

Jeremy Wolfe (signed review)

---

## [Decision Letter · Decision Letter 1]

28 Nov 2023

Just Say ‘I Don’t Know’: Understanding Information Stagnation during a Highly Ambiguous Visual Search Task

PONE-D-23-15409R1

Dear Dr. Godwin,

We’re pleased to inform you that your manuscript has been judged scientifically suitable for publication and will be formally accepted for publication once it meets all outstanding technical requirements.

Kind regards,

Sathishkumar Veerappampalayam Easwaramoorthy

Academic Editor

PLOS ONE

Additional Editor Comments (optional):

Reviewers' comments:

Reviewer's Responses to Questions

**Comments to the Author**

1. If the authors have adequately addressed your comments raised in a previous round of review and you feel that this manuscript is now acceptable for publication, you may indicate that here to bypass the “Comments to the Author” section, enter your conflict of interest statement in the “Confidential to Editor” section, and submit your "Accept" recommendation.

Reviewer #1: All comments have been addressed

Reviewer #2: All comments have been addressed

2. Is the manuscript technically sound, and do the data support the conclusions?

Reviewer #1: Yes

Reviewer #2: Yes

3. Has the statistical analysis been performed appropriately and rigorously? 

Reviewer #1: Yes

Reviewer #2: Yes

4. Have the authors made all data underlying the findings in their manuscript fully available?

Reviewer #1: Yes

Reviewer #2: Yes

5. Is the manuscript presented in an intelligible fashion and written in standard English?

Reviewer #1: Yes

Reviewer #2: Yes

6. Review Comments to the Author

Reviewer #1: I thank the authors for their diligent revision of their paper and their responses to my comments. I believe that this study can be published in PLOS ONE.

Reviewer #2: The authors have done a thorough job in addressing my comments and I think this will make a nice contribution to the literature. It'll be interesting to see how the future experiments pan out.

7. PLOS authors have the option to publish the peer review history of their article (what does this mean?). If published, this will include your full peer review and any attached files.

Reviewer #1: No

Reviewer #2: No

---

## [Editor Report · Acceptance letter]

29 Nov 2023

PONE-D-23-15409R1 

Just Say ‘I Don’t Know’: Understanding Information Stagnation during a Highly Ambiguous Visual Search Task 

Dear Dr. Godwin:

I'm pleased to inform you that your manuscript has been deemed suitable for publication in PLOS ONE. Congratulations! Your manuscript is now with our production department. 

Kind regards, 

on behalf of

Dr. Sathishkumar Veerappampalayam Easwaramoorthy 

Academic Editor

PLOS ONE